

**Development and Preliminary Testing of Temporally**
**Controllable Weather Modification Rocket with Spatial Seeding**
**Capacity**
Dong Xiaobo[1, 2, 3], Wang Xiaoqing[3], Liu Yongde[4], Wang Xiaorong[4]
[1]China Meteorological Administration Xiong'an Atmospheric Boundary Layer Key Laboratory, Xiong'an New
Area, 071800, China
[2]Key Laboratory of Meteorology and Ecological Environment of Hebei Province, Shijiazhuang, 050021, China
[3]Hebei Provincial Weather Modification Center, Shijiazhuang, 050020, China
[4]Xi'an Qinghua Commercial Explosives Corp., Ltd., Xi'an, 710025, China
*Correspondence to*: Dong Xiaobo (xb.dong@qq.com)
**Abstract.** Current weather modification rockets with a single operation mode, limited operating height, and fixed
and uncontrollable operating time, cannot achieve seeding in different layers, stages and quantities for different
cloud systems. Therefore, a temporally controllable weather modification rocket with spatial seeding capability is
developed in this study. This new rocket features an electronic fuse-controlled intelligent ignition system, with eight
channels of ignition tube outputs. Additionally, carrier wave communication technology is incorporated to set the
seeding time for eight sets of ignition tubes. The temporally controllable rocket is capable of initiating seeding
within 2 s to 26 s and can conduct operations for layering, arbitrary altitude and fractional-dose seeding within the
altitude range of 500–5500 m (at a launch angle of 70°). The minimum time interval of the rocket for seeding can
be set to 0.1 s, and all 48 catalyst bullets loaded in a single rocket can be launched within 0.8 s. Thus, the rocket can
achieve both concentrated and continuous seeding. Consequently, during weather modification operations,
parameters such as altitude, thickness and operating temperature of target clouds can be obtained through detection,
and they can be used to automatically calculate the suitable seeding time, seeding altitude and seeding dose in order
to improve the accuracy and scientificity of cloud catalytic operations. Ground tests show that the reliabilities of the
electric ignition tube output, new electronic fuse input and output, and electronic fuse output energy all meet the
design requirements. The temporally controllable spatial-seeding rocket can achieve adjustable and controllable
seeding times for catalytic bullets, meeting the safety and reliability requirements of rockets.
**Keywords:** Temporally controllable; Weather modification; Spatial seeding rockets; Cloud seeding



**1 Introduction**

Weather modification primarily involves the dispersal of a certain amount of seeding agents such as silver iodide and dry ice into clouds at appropriate locations under conducive weather conditions using various tools such as aircraft, anti-aircraft guns, rockets and ground generators, in order to achieve artificial rainfall (snow), hail suppression and fog dispersal (Bruintjes, 1999; Mao and Zheng, 2006). Currently, silver iodide is the main catalyst used for cold cloud seeding operations. The quantity of effective ice nuclei produced during cold cloud seeding is related to the ambient temperature. Experimental results have demonstrated that the nucleation rate is the highest when the temperature of clouds ranges from −20 °C to −4 °C. Moreover, the ideal temperature range is between −12 °C and −5 °C, where silver iodide exhibits the optimal ice nucleation effects (Lou et al., 2021).

With advancements in cloud and precipitation detection technology, research on the structure of target clouds for weather modification has become increasingly sophisticated, leading to improved accuracy in identifying operational conditions and locations. Aircraft-based precipitation enhancement operations can penetrate stratiform clouds, detect cloud microphysical characteristics, identify suitable seeding areas and conduct scientifically precise operations (Guo et al., 2021). However, during severe convective weather, aircraft operations are not feasible. Moreover, for widespread stratiform precipitating cloud systems, aircraft operations are constrained by factors such as flight range, flight time, and seeding dosage, limiting coverage of the entire operational area. Ground-based weather modification rocket operations can circumvent threats posed by high-altitude conditions and are suitable for situations where aircraft operations are challenging. Additionally, rocket operations in weather modification can complement aircraft operations to expand seeding coverage and improve seeding efficiency.

In northern China, the microphysical mechanisms of catalyzing stratiform clouds for artificial rainfall mainly involve the coordination of seeding clouds in the upper levels and feeder clouds in the lower levels to form precipitation. Seeding and feeder clouds are often governed by different dynamic conditions, and sometimes, they are not continuous cloud bodies, with occasional dry layers of different thicknesses between them (Yao, 2006; Hong and Lei, 2012; Dong et al., 2022). In hail suppression operations, it is desired that seeding agents be concentrated near the "funnel" of hail clouds (Xu, 2001; Xu and Tian, 2008; Yao et al., 2022). Observation results of stratiform clouds for winter snowfall in northern China from 2017 to 2022 indicated that winter stratiform clouds for snowfall exhibit more stability in development than spring and autumn stratiform precipitating clouds, with less fluctuations in cloud top height (mostly lower than 3500 m) (Dong et al., 2020, 2021; Liu et al., 2021; Fu et al., 2023; Wang et al., 2023; Yan et al., 2023). In operations, the seeding height of most weather modification rockets is within 3–6





km.
Currently, rocket seeding techniques in weather modification are categorized into line seeding and spatial seeding
modes (Wang et al., 2018). Line seeding rockets have fixed seeding start times set at the time of production, and the
seeding height varies mainly with the launch angles of the rocket launcher, with a relatively limited adjustable
range of seeding altitudes. For spatial seeding rockets, although the start seeding time can be set before launch, the
adjustment range is currently limited to 3–12 seconds. With a launch angle of 55°, the start seeding height is
approximately between 2000 m and 3200 m, and seeding operations cannot be interrupted after starting. These two
techniques carry out seeding operations on variable clouds with fixed settings in rockets. Thus, there are several
phenomena, i.e., the temperature in the seeding position does not reach the effective temperature range of silver
iodide, the seeding occurs in dry layers of stratiform clouds, and there is empty seeding (the seeding location is
above cloud tops in winter).
To address the above challenges, this study aims to develop a new generation of weather modification rockets with
temporally controllable and adjustable seeding capabilities. By presetting accurate seeding start times, quantities
and end times before rocket launch based on the actual height and thickness of clouds at the operational site, these
rockets can enter target clouds and conduct operations according to the preset seeding parameters, thereby
improving the scientific and the precision of weather modification rockets for precipitation enhancement and hail
suppression.
The remainder of this paper is arranged as follows. Section 2 introduces the principles of spatial-seeding rockets for
weather modification. Section 3 discusses the principles of the spatial-seeding rocket with a temporally controllable
ability. The ground testing results for temporally controllable spatial-seeding rockets are analyzed in sec. 4. Section
5 investigates the safety of temporally controllable spatial-seeding rockets. The main conclusions are presented in
sect. 6.
**2. Principles of spatial-seeding rockets for weather modification**
The technology of spatial-seeding rockets for weather modification is derived from the concept of submunition, i.e.,
the internal ejection mechanism generates power to disperse submunitions from cluster munition upon reaching the
target, thus improving the utilization efficiency of ammunition. For instance, the projectile of the ZBZ-HJ-7
spatial-seeding rocket consists of a dispenser, a safety landing system (parachute cabin), an engine and a tail fin
(Fig. 1). The rocket has a diameter of 82 mm, a length of 1660 mm and a weight of 10.8 kg.

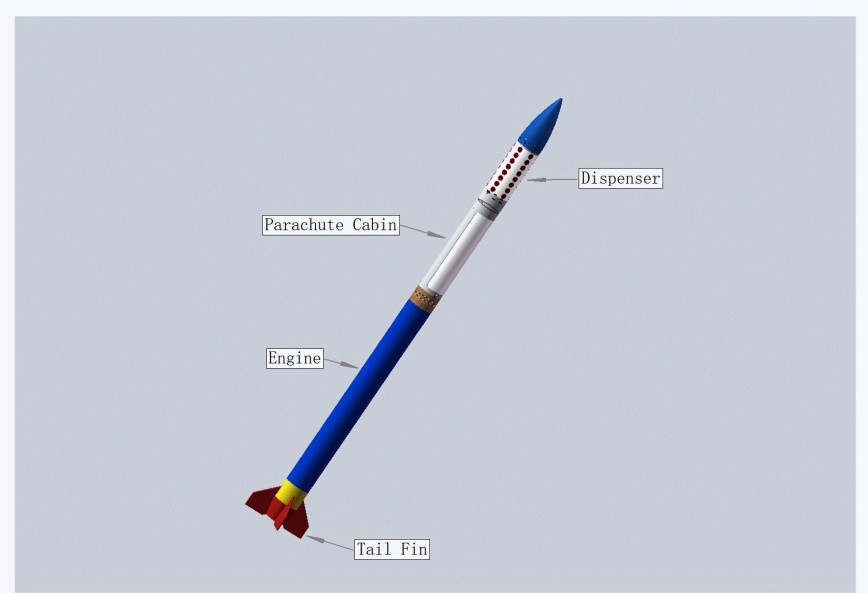


**Figure 1: Schematic diagram of the ZBZ-HJ-7 spatial-seeding rocket.**


The dispenser consists of one wind cap, eight sets of launch modules, one control cabin and one electronic fuse.
Each set of launch modules consists of an upper cover, a lower cover, six bullets filled with catalysts, and one delay
ignition tube. The structure diagram of a single launch module is shown in Fig. 2.

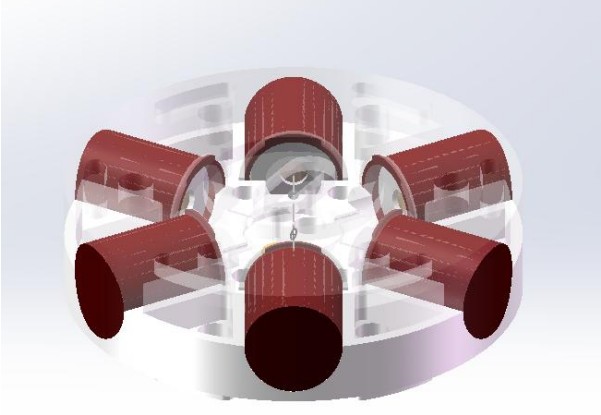


92                      **Figure 2: Structure diagram of a single launch module.**


The main difference between spatial-seeding rockets and line-seeding rockets lies in the structure of the warhead
charge. For line-seeding rockets, a catalyst charge column is assembled in the warhead. When the charge column is





ignited from the end face, the combustion products are dispersed outward along the trajectory through small holes
on the side of the projectile body. In contrast, in terms of spatial-seeding rockets, the catalytic material is packed
into the bullets, which are vertically mounted along the projectile axis inside the warhead cylinder. A delay ignition
tube matching the catalyst bullets is installed in the center of the projectile axis. After the rocket is launched, the
delay ignition tube will be ignited at the setting time, thereby igniting the surrounding catalyst bullets. Then, the
propellant inside the catalyst bullets will propel them out of the projectile body and ignite the catalyst inside the
bullets simultaneously. The high temperature generated by combustion causes silver iodide crystals and other
substances in the catalyst to sublime into silver iodide aerosol.
The ZBZ-HJ-7 spatial-seeding rocket carries a total of 48 catalyst bullets, arranged in eight rows along the rocket
axis, with six bullets per row. After the rocket is launched, the electronic fuse starts timing, and after reaching the
seeding time (preset before launching), the electronic fuse ignites the delay ignition tube in the first set of launch
modules. After a delay of 2.7 s, the delay ignition tube ignites the six catalyst bullets in the first set of launch
modules, which exit the cabin at a speed of $\geq 40$ m·s$^{-1}$. Simultaneously, the delay ignition tube in the next set of
launch modules is ignited. After another delay of 2.7 s, the six catalyst bullets in the second set of launch modules
are ignited. In this way, all eight sets of launch modules are fired. Near the apex of the flight trajectory, the
spatial-seeding rocket deploys a parachute, and the debris descends slowly. Each catalyst bullet carries 6.2 g of
silver iodide. The combustion of the catalyst bullets forms a silver iodide aerosol band in the space they traverse.
The flight trajectory of the bullets is perpendicular to that of the rocket, with a flight distance of at least 200 m.
After all the bullets for precipitation enhancement are dispersed from the projectile body, an instantaneous catalytic
zone with a diffusion cross-section diameter of approximately 500 m and a length of 6.6 km is formed in space.
**3. Principles of temporally controllable spatial-seeding rockets**
To achieve precise seeding for target clouds at different heights and thicknesses, as well as multi-layer cloud
systems, a comprehensive analysis and design improvement are conducted on the ZBZ-HJ-7 spatial-seeding rocket.
The aim is to ensure that each layer of launch modules disperses according to the flight trajectory (time), replacing
the original design scheme of fixed-delay ignition tubes with a new sequential ignition design controlled by
electronic fuses. Seven ignition outputs are added to this new design scheme. The start seeding time of the rocket
can be set within 2–26 s, corresponding to a range of start seeding heights within 500–5,500 m (at a launch angle of
70°). The minimum seeding time interval can be set to 0.1 s, allowing all 48 catalyst bullets carried by a single
rocket to be launched within 0.8 s. To realize these improvements, four designs are made for the rocket. Firstly,
electric ignition tubes are installed in the launch modules, requiring structural and circuitry redesign. Secondly, the
electronic fuse outputs increase from one to eight, and the electrical interface is redesigned (Fig. 3). Additionally,
changes are made to the parameters received by the electronic fuse and the data output, requiring the redesign of
parameter binding methods and software. Finally, the functions of the electronic fuse and the dispenser are largely
altered, and the testing and inspection methods are redesigned.

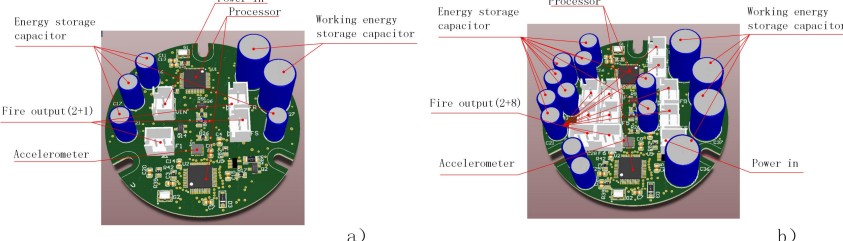

**Figure 3: Three-dimensional model diagrams of the electronic fuses of (a) the ZBZ-HJ-7 spatial-seeding rocket and (b)**
**the temporally controllable spatial-seeding rocket.**

The operational design of temporally controllable spatial-seeding rockets is outlined as follows. Before weather
modification operations, the multi-source observation data are integrated and analyzed to obtain the parameters
suitable to target clouds, such as the number of cloud layers, cloud base height and cloud thickness. These
parameters are input into the launch control system along with launch parameters (local altitude and launch angle).
Then, the launch control system automatically calculates the seeding time of eight sets for target clouds based on
the flight trajectory of rockets (Fig. 4a). Afterwards, the seeding time of eight sets is automatically loaded into
rockets by using carrier wave communication technology. The two contacts on rockets are connected to the launch
control system through wires on the launching rack. The launch control system applies electrical signals with a
specific frequency to the two contacts, which are demodulated by the electronic fuse to set the seeding times
correctly.
The rocket begins timing from the moment of liftoff. At each set seeding time point, six catalyst bullets in each row
are ejected. For example, assume that the target cloud system has two layers, with the lower layer in heights of
1000–3000 m and the upper layer in heights of 4000–4500 m. As the lower layer is thicker and the upper layer is
thinner, the seeding dosage can be set to five rows of bullets for the lower layer and three rows for the upper layer.
If the launch angle is 65°, the theoretical calculation indicates that the seeding operation of the rocket begins after
2.6 s in the first layer, at an approximate height of 1,200 m. The time interval between the ejections of each set of





bullets is 1 s, and the seeding is completed after 5 s (Fig. 4b), at which time the altitude of the rocket is
approximately 2800 m. The dispersion in the second layer begins after 12.3 s, at an altitude of approximately 4,200
m. Since the thickness of the cloud layer is only 500 m, and the flight speed of the rocket decreases when reaching a
high altitude, the seeding interval can be set to 0.2 s, thus achieving the goal of concentrated seeding within a
thinner cloud layer (Fig. 4c).

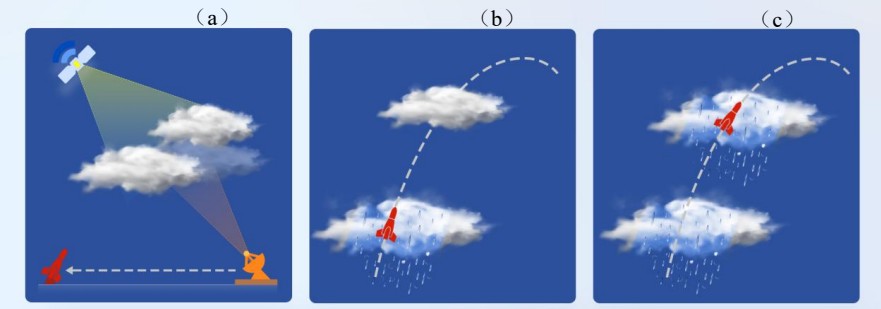

**Figure 4: Schematic diagram of the working demo of a temporally controllable spatial-seeding rocket.**
**4. Ground testing result analysis for a temporally controllable spatial seeding rocket**
A temporally controllable spatial-seeding rocket is deeply improved and developed on the basis of the ZBZ-HJ-7
spatial-seeding rocket. According to the requirements of engineering design, it is necessary to conduct
comprehensive ground tests and experiments to verify the feasibility and reliability of the new technical state. The
ground tests primarily consist of three main parts: reliability tests of the electric ignition tube outputs, the new
electronic fuse input and output, and the output energy of the new electronic fuse.
**4.1 Reliability test of electric ignition tube outputs**
One major improvement in temporally controllable spatial-seeding rockets is the replacement of the electric ignition
tubes with the original ignition tube with a fixed delay of 2.7 s. The electric ignition tubes mainly consist of a
bridge wire, igniting powder and lead wires. The igniting powder tightly encases the bridge wire, and when current
passes through the bridge wire, heat is generated according to Joule's law ($Q = I^2 \cdot R \cdot t$). The bridge wire is highly
sensitive to heat, and when heat is generated, the temperature of the bridge wire instantly rises. When the
temperature reaches the ignition temperature of igniting powder, igniting powder ignites, producing
high-temperature gas and hot metal particles to ignite the initiating explosive device of the next stage. Temporally
controllable spatial-seeding rockets prioritize the use of matured standardized No. 32 electric ignition tubes as





ignition components. These tubes have an ignition compound weight of 18–22 mg, ensuring stable and reliable
ignition outputs.
In order to test the reliability of electric ignition tubes to ignite catalyst bullets in the launch module, a dispenser
assembled with eight sets of launch modules, each equipped with one electric ignition tube, is designed. One set of
electric ignition tubes is randomly selected to input igniting energy, and the ignition status of catalyst bullets in this
set is observed. A total of eight sets of launch modules, namely eight electric ignition tubes, are tested, igniting a
total of 48 catalyst bullets. The test results show that after the electric ignition tubes work, all six catalyst bullets in
each set of launch modules are ignited. The catalyst bullets fly out of the launch modules at high speed and burn
stably during the flight, and their burning time meets the technical specifications. The proper functioning of electric
ignition tubes determines whether the dispenser can operate effectively, which is critical in the entire ignition. To
ensure the reliability of the electric ignition tubes to ignite catalyst bullets, in the formal design of the dispenser,
two electric ignition tubes are installed in each set of launch modules as backups for each other. Hence, tests are
conducted with eight sets of launch modules, each equipped with two electric ignition tubes, totaling 16 electric
ignition tubes and 48 catalyst bullets. All tested catalyst bullets fly and burn normally after the electric ignition
tubes are activated.
**4.2 Reliability test of the input and output of the new electronic fuses**
To test the accuracy and reliability of the output timing sequence of the new electronic fuse, a dedicated upper
computer software for the electronic fuse is designed. A seeding time calculation program is developed, and an
output signal acquisition system is constructed. The testing methods are as follows. Firstly, the cloud parameters
and launch angles are input into the seeding time calculation program to calculate eight sets of seeding time points
and the optimal parachute opening time. Subsequently, the eight sets of seeding times and parachute opening times
are loaded into the electronic fuse through the dedicated upper computer. Finally, the signal acquisition system is
docked with the electronic fuse. After the electronic fuse is powered on, the signal acquisition system automatically
records the output time of the timing controller. Table 1 shows the test results of one of the electronic fuses.
After repeated tests on ten electronic fuses, the cumulative valid data records are over 4,800. The test results show
that the deviation between the actual output time and the loaded time is within ±0.1 s, indicating that the electronic
fuses output accurate seeding time points and are reliable.

**Table 1. Test results of an electronic fuse for seeding time: loaded and actual output.**

| Serial number | | Parachute opening time (s) | Seeding time (s) | | | | | | | | |
|---|---|---|---|---|---|---|---|---|---|---|---|
| 1 | Load time | 28.9 | 3.1 | 3.8 | 4.6 | 5.3 | 10.4 | 12.3 | 14.2 | 16.1 | |
| | Observed time | 28.8 | 2.9 | 3.7 | 4.4 | 5.1 | 10.2 | 12.1 | 14.1 | 15.9 | |
| | | 28.8 | 2.9 | 3.7 | 4.4 | 5.1 | 10.3 | 12.1 | 14.1 | 15.9 | |
| | | 28.9 | 3.2 | 3.8 | 4.5 | 5.2 | 10.5 | 12.2 | 14.2 | 16.0 | |
| | | 29.1 | 3.1 | 3.8 | 4.6 | 5.3 | 10.4 | 12.4 | 14.2 | 16.1 | |
| 2 | Load time | 30.4 | 64 | 103 | 142 | 181 | 220 | 259 | 289 | 337 | |
| | Observed time | 30.2 | 63.9 | 102.8 | 141.9 | 180.9 | 220 | 258.8 | 288.8 | 336.8 | |
| | | 30.3 | 63.9 | 102.8 | 141.8 | 108.8 | 218.8 | 258.8 | 288.9 | 337.2 | |
| | | 30.4 | 63.9 | 102.9 | 141.9 | 180.9 | 220 | 259 | 289.0 | 336.9 | |
| | | 30.3 | 63.9 | 102.9 | 141.8 | 180.9 | 219.9 | 258.9 | 289.2 | 336.9 | |
| 3 | Load time | 50 | 100 | 150 | 200 | 250 | 300 | 350 | 400 | 450 | |
| | Observed time | 49.8 | 99.8 | 149.9 | 199.9 | 249.8 | 299.8 | 349.9 | 399.8 | 449.8 | |

Note: Serial number 3 indicates the operational capability of the timing controller under extreme conditions.

**4.3 Reliability test of the output energy of the new electronic fuses**

The prerequisite for electric ignition tubes to ignite is that the energy input by electronic fuses must be greater than the critical igniting energy of electric ignition tubes. In other words, the output energy of electronic fuses determines the reliability of igniting electric ignition tubes. The most efficient way for the electronic fuses to output igniting energy is by discharging the capacitor ( $Q = 0.5 \cdot C \cdot U^2$ , where Q denotes the output energy, C the capacitance, and U the voltage across the capacitor). In engineering practice, a fixed capacitance capacity is often used, and the voltage across the capacitor is adjusted to test whether the output energy can ignite the corresponding initiating explosive devices.

In this research, in order to verify whether the new electronic fuses can reliably ignite the electric ignition tubes in the launch modules, a set of sensitivity tests is conducted on the new electronic fuses following the method specified in *The up-and-down estimate method for sensitivity test* (Standard No.: GJB 377–1987). With the capacitance kept constant, a set of sensitivity tests is performed with the capacitor discharge voltage as the variable. Based on the 32 valid data accumulated in the test and the equation $Q = 0.5 \cdot C \cdot U^2$ , we can calculate that the igniting reliability can achieve 99.99% when the capacitance capacity (C) is 220 μF and the capacitor voltage (U) is 11 V, which meets the critical energy for reliable ignition of electric ignition tubes.





Subsequently, the new electronic fuse undergoes a test for its maximum operating time. At its maximum possible
operating time, the output capacitor voltage is measured not to be lower than 16 V, suggesting that the actual output
energy of electronic fuses is approximately 2.1 times higher than the critical igniting energy for electric ignition
tubes (with a capacitor voltage U of 11 V), which meets the general design requirements for a pyrotechnic sequence
with a 2-fold ignition margin.
After the output igniting energy of new electronic fuses is decided to be sufficient, an output reliability test for new
electronic fuses is conducted. Each new electronic fuse initiates eight sets of dual electric ignition tubes, and a total
of 80 tubes are ignited by ten electronic fuses. All electric ignition tubes ignite normally, and the ignition time
points are consistent with the loaded time points. The test results suggest that the electronic fuses can reliably ignite
electric ignition tubes in ground tests.
**4.4 Ground tests of dispenser**
Based on the tests of individual subsystems above, two sets of dispensers are assembled for ground seeding and
flight distance testing of catalyst bullets in formal operational status to assess the overall performance of the
improved dispensers. The two sets of dispensers are powered on and initiated according to the load times of serial
numbers 1 and 2 in Table 1. The results reveal that all 96 catalyst bullets are launched from the launchers at the
preset time. The catalyst bullets burn normally, and the flight distances meet the technical requirements. These
findings indicate that improved temporally controllable spatial-seeding rockets meet the design requirements and
achieve controllable and accurate seeding.
**5. Safety of temporally controllable spatial seeding rockets**
Based on the ZBZ-HJ-7 rocket, a temporally controllable spatial-seeding rocket improves the igniting sequence of
the dispenser. Since the structure of the rocket body is the same as that of the original rocket, the trajectory and
safety program also remain unchanged. The main technical specifications of temporally controllable rockets and the
ZBZ-HJ-7 spatial-seeding rocket are listed in Table 2.

**Table 2. Main technical specifications of the temporally controllable spatial-seeding rockets and the**
**ZBZ-HJ-7 spatial-seeding rocket.**

| Technical specifications | ZBZ-HJ-7 rocket | Temporally controllable rocket |
|---|---|---|
| Rocket diameter (mm) | 82 | 82 |
| Bullet length (mm) | 1660 | 1660 |
| Bullet weight (kg) | 10.8 | 10.8 |





| Maximum launch height (m) | ≥7000 (85°) | ≥7000 (85°) |
| --- | --- | --- |
| Number of catalyst bullets (rounds) | 48 | 48 |
| Total amount of silver iodide carried by catalyst bullets (g) | ≥20 | ≥20 |
| Nucleation rate of silver iodide at −10°C (pcs/g) | ≥$1.18 \times 10^{14}$ | ≥$1.18 \times 10^{14}$ |
| Lateral flight speed of single bullets (m/s) | ≥40 | ≥40 |
| Burning time of single bullets (s) | 6–9 | 6–9 |
| Seeding time of the total bullets (s) | ≥27 | 0.8–32 |
| Load time range of electronic fuses (s) | 6–17 | 2–26 |
| Seeding time interval of electronic fuses (s) | 2.7 | 0.1–26 |
| Landing speed of debris (m/s) | ≤8 | ≤8 |


During the long storage validity period, the rocket body is sealed, and the internal propellant is designed to
withstand the environment within −30°C to 50°C. Temperature and humidity variations can not decrease the safety
of rocket body. After loading onto the launching rack and before being powered on, the internal initiating explosive
devices of rockets are all semi-insensitive, with a safety current level higher than that of common civilian initiating
explosive devices, which can withstand general stray current shocks and possess a higher ability to resist accidental
electrical stimulation. During the self-check and loading phase after powering on, the voltage in the electronic fuses
is at only 3.3 V, which even under extreme conditions does not trigger the internal ignition sequence. Upon pressing
the launch button, the electronic fuses charge to a working voltage of 24 V within 1 s. Once detecting that the
capacitor voltage exceeds the critical value, the electronic fuses immediately ignite the engine. Subsequently, the
rocket leaves the launching rack at high speed, and electronic fuses start timing. Any lapse or failure in the
aforementioned process prompts an immediate internal discharge program. In an extreme case, if the engine ignites
but the propellant fails to burn, the rocket does not leave the launching rack and the attitude sensor in the electronic
fuse does not receive a flight signal. In this situation, all the energy stored in the capacitors of electronic fuses will
be discharged within 3 s to ensure that the rocket will not seed or deploy its parachute on the launching rack, thus
remaining in a safe and unpowered state for subsequent handling.
The new design sets the operation time for rockets by using carrier wave communication technology. The input
signal must pass through a demodulation program, and until the input signal is demodulated, the electronic fuses





consider any input signals as invalid. Directly applying launch energy or other forms of energy, electronic fuses will
not activate the working program. This design eliminates the safety risk of the rocket launching as soon as it is
powered on. The parachute deployment mechanism, which reliably operates near the apex of the trajectory, has
been used over 50,000 instances in the ZBZ-HJ-7 rocket, demonstrating its reliability.
**6. Conclusions**
In order to achieve precise, timed and quantified seeding operations of weather modification rockets, the temporally
controllable spatial-seeding rocket is developed by redesigning the ignition control unit and improving the
operational parameters loading method. During seeding operations, parameters such as cloud height, thickness and
operating temperature can be automatically calculated by using detected data. Based on these parameters, the
appropriate seeding time points, heights and dosages can be determined, thus enhancing the precision and
scientificity of cloud seeding operations. The main conclusions are as follows.
The newly designed temporally controllable spatial-seeding rocket replaces the fixed-delay ignition with intelligent
ignition controlled by electronic fuses, with eight-channel electric ignition tubes. Additionally, carrier wave
communication technology is used to set operational parameters, which enable the rocket to have functions such as
adjustable and controllable time sequences for seeding operations. The seeding start time of rockets can be set as
2–26 s, and the seeding height varies in the range of 500–5500 m (at a launch angle of 70°). Within this range,
seeding operations can be carried out at different altitudes and dosages. The minimum seeding time interval can be
set to 0.1 s, enabling all 48 catalyst bullets to be launched within 0.8 s, thereby achieving both continuous and
concentrated seeding.
Ground tests are conducted on the temporally controllable spatial-seeding rocket, including reliability tests of
electric ignition tube output, new electronic fuse input and output, and electronic fuse output energy. The results
indicate that all 16 electric ignition tubes work reliably, and 48 catalyst bullets launch and ignite normally. Over
4,800 valid data points are collected during repeated testing of ten electronic fuses. The deviation between the
actual output time of the electronic fuses and the set time is no more than ±0.1 s. Additionally, all 80 electric
ignition tubes are successfully ignited. Based on preliminary testing, ground tests are conducted on the dispenser.
All the 96 catalyst bullets are dispersed according to the preset time and burn normally. The improved temporally
controllable spatial-seeding rocket meets design requirements, achieving controllable and precise seeding.
The temporally controllable spatial-seeding rocket is based on the structure of the ZBZ-HJ-7 spatial-seeding rocket,



with improvements limited to the dispenser, while the engine and parachute compartment remain unchanged. In
addition, the aerodynamic shape, mass center and thrust of the rocket remain unaltered, and thus, the ballistic
trajectory is the same as that of the original design. The use of carrier wave communication technology eliminates
the safety risk of rocket launching as soon as it is powered on, ensuring the safety and reliability of the rocket.
Based on the insights in this study, future research will involve observation and catalyst experimental design and
carry out catalyst tests for different precipitating cloud systems in northern China by using temporally controllable
spatial-seeding rockets. Moreover, Experiment data will be accumulated, and physical tests of the effects of cold
cloud seeding operations will be conducted to verify the actual effects of the new seeding rocket.

***Author contributions***
Dong xiaobo performed the experiments designed by Wang Xiaoqing and Liu Yongde. Dong Xiaobo wrote the first
draft, which was further revised by Liu Yongde, and Wang Xiaoqing.
***Competing interests***
The authors declare that they have no conflict of interest.
***Disclaimer***
Publisher's note: Copernicus Publications remains neutral with regard to jurisdictional claims made in the text,
published maps, institutional affiliations, or any other geographical representation in this paper. While Copernicus
Publications makes every effort to include appropriate place names, the final responsibility lies with the authors.
***Acknowledgement***
This work was supported by Hebei Natural Science Foundation (D2023304001) and China Meteorological
Administration Innovation and Development Project(CXFZ2023J038).We thank Nanjing Hurricane Translation for
reviewing the English language quality of this paper.

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
