# Peer review of "Development and Preliminary Testing of Temporally"

_Atmospheric Measurement Techniques, 2024_

## Author Response (AR2)

**Reponse**

Dear respected editor and reviewers,

Thank you very much for your valuable suggestions that will help improve the quality of this paper. We have revised our manuscript according to your comments, and the corrections are detailedly listed below.

RC1:

This research redesign a new Weather Modification Rocket with Spatial Seeding Capacity. The new rocket can achieve both concentrated and continuous seeding. Consequently, during weather modification operations, parameters such as altitude, thickness and operating temperature of target clouds can be obtained through detection, and they can be used to automatically calculate the suitable seeding time, seeding altitude and seeding dose in order to improve the accuracy and scientificity of cloud catalytic operations.This paper is overall well-written, I think minor revisions are needed before acceptance. Below listed are the comments and suggestions.

**Response to Reviewer 1**: Thanks to your comments and suggestions. Following your suggestions, we have revised the manuscript. The point-by-point response is listed below according to your specific comments.

1. Line 18: " altitude range of 500 – 5500 m (at a launch angle of 70°). " Is this altitude range obtained through actual measurement or calculation?

Reply:The temporally controllable rocket is capable of initiating seeding within 2–26 s, and can conduct operations for layering, arbitrary altitude and fractional-dose seeding within the altitude range of 500–5500 m (at a launch angle of 70°), which is calculated based on the flight trajectory of rockets.

2. Line 145: "The rocket begins timing from the moment of liftoff. " should be liftoff the launching rack.

Reply:Thanks for your valuable suggestions. The rocket begins timing from the moment of liftoff the launching rack.

3. Table 2: Please explain the differences between the two different rockets.

Reply:Thanks for your valuable suggestions. We add a paragraph on line 239.

"The seeding time of the total bullets has been changed from ≥ 27 seconds to 0.8–32 seconds, which reflects the adjustable and controllable ejection height and duration of the temporally controllable spatial-seeding rocket. The load time range of electronic fuses has been changed from 6–17 s to 2–26 s, expanding the range of new rocket seeding heights. The seeding time interval of electronic fuses has been changed from the

original fixed 2.7 s to 0.1–26 s, which can achieve continuous and concentrated seeding."

4. Line 298: Author contributions didn't give the role of the fourth author.

Reply:Thanks for your valuable suggestions. We have revised this part. The author Dong Xiaobo provided methods and ideas for the temporally controllable spatial-seeding rocket. The experiments were designed by Liu Yongde and Wang Xiaorong. Dong Xiaobo wrote the first draft, which was further revised by Liu Yongde and Wang Xiaoqing.

RC2:

**Response to Reviewer 2**: Thanks to your comments and suggestions. Following your suggestions, we have revised the manuscript. The point-by-point response is listed below according to your specific comments.

1. Line 21: " parameters such as altitude, thickness and operating temperature of target clouds can be obtained through detection, and they can be used to automatically calculate the suitable seeding time, seeding altitude and seeding dose".Is it calculated automatically by the the launch control system?

Reply : The parameters such as altitude, thickness and operating temperature of target clouds can be obtained through detection, and they

can be used to automatically calculate the suitable seeding time, seeding altitude and seeding dose by the launch control system based on the flight trajectory of rockets.

2. Line 91: Please mark the upper cover, lower cover, bullets in the Fig2.

Reply:Thanks for your valuable suggestions. We have revised Figure 2.

3. Line 108:"Simultaneously, the delay ignition tube in the next set of launch modules is ignited. "Should be the second set of launch modules is ignited.

Reply:Simultaneously, the delay ignition tube in the second set of launch modules is ignited.

4. Please provide a more detailed explanation of the characteristics of line seeding rockets.

Reply:Thanks for your valuable suggestions. We have added the more detailed explanation for the line seeding rockets on line 61.

"At present, the commonly used line seeding rocket has a seeding time of 15–30 seconds and a maximum launch altitude of 6–8 kilometers. After the rocket is launched, the catalyst burns inside the rocket body and spreads along the direction of the rocket's flight trajectory. After seeding, a long linear silver iodide aerosol band is formed in the air."

RC3:

Adjust the author list as follows: First name followed by last name.

Reply:I have adjusted the author list.